# The Value of APTw CEST MRI in Routine Clinical Assessment of Human Brain Tumor Patients at 3T

**DOI:** 10.3390/diagnostics12020490

**Published:** 2022-02-14

**Authors:** Julia P. Lingl, Arthur Wunderlich, Steffen Goerke, Daniel Paech, Mark E. Ladd, Patrick Liebig, Andrej Pala, Soung Yung Kim, Michael Braun, Bernd L. Schmitz, Meinrad Beer, Johannes Rosskopf

**Affiliations:** 1Department of Radiology, Ulm University, Albert-Einstein-Allee 23, 89081 Ulm, Germany; julia.lingl@uni-ulm.de (J.P.L.); arthur.wunderlich@uni-ulm.de (A.W.); sykim1980@hotmail.de (S.Y.K.); michael-1.braun@uni-ulm.de (M.B.); bernd.schmitz@uni-ulm.de (B.L.S.); meinrad.beer@uniklinik-ulm.de (M.B.); 2German Cancer Research Center (DKFZ), Department of Medical Physics in Radiology, Im Neuenheimer Feld 280, 69120 Heidelberg, Germany; s.goerke@dkfz-heidelberg.de (S.G.); mark.ladd@dkfz-heidelberg.de (M.E.L.); 3German Cancer Research Center (DKFZ), Division of Radiology, Im Neuenheimer Feld 280, 69120 Heidelberg, Germany; d.paech@dkfz-heidelberg.de; 4Department of Neuroradiology, Venusberg-Campus 1, Bonn University, 53127 Bonn, Germany; 5Faculty of Medicine, University of Heidelberg, Im Neuenheimer Feld 672, 69120 Heidelberg, Germany; 6Faculty of Physics and Astronomy, University of Heidelberg, Im Neuenheimer Feld 226, 69120 Heidelberg, Germany; 7Siemens Healthcare GmbH, Henkestraße 127, 91052 Erlangen, Germany; patrick.liebig@siemens-healthineers.com; 8Department of Neurosurgery, Bezirkskrankenhaus Guenzburg, Lindenallee 2, 89312 Guenzburg, Germany; andrej.pala@uni-ulm.de; 9Section of Neuroradiology, Bezirkskrankenhaus Guenzburg, Lindenallee 2, 89312 Guenzburg, Germany

**Keywords:** CEST, APTw, brain tumor, MRI

## Abstract

Background. With fast-growing evidence in literature for clinical applications of chemical exchange saturation transfer (CEST) magnetic resonance imaging (MRI), this prospective study aimed at applying amide proton transfer-weighted (APTw) CEST imaging in a clinical setting to assess its diagnostic potential in differentiation of intracranial tumors at 3 tesla (T). Methods. Using the asymmetry magnetization transfer ratio (MTRasym) analysis, CEST signals were quantitatively investigated in the tumor areas and in a similar sized region of the normal-appearing white matter (NAWM) on the contralateral hemisphere of 27 patients with intracranial tumors. Area under curve (AUC) analyses were used and results were compared to perfusion-weighted imaging (PWI). Results. Using APTw CEST, contrast-enhancing tumor areas showed significantly higher APTw CEST metrics than contralateral NAWM (AUC = 0.82; *p* < 0.01). In subgroup analyses of each tumor entity vs. NAWM, statistically significant effects were yielded for glioblastomas (AUC = 0.96; *p* < 0.01) and for meningiomas (AUC = 1.0; *p* < 0.01) but not for lymphomas as well as metastases (*p* > 0.05). PWI showed results comparable to APTw CEST in glioblastoma (*p* < 0.01). Conclusions. This prospective study confirmed the high diagnostic potential of APTw CEST imaging in a routine clinical setting to differentiate brain tumors.

## 1. Introduction

Intracranial tumors exhibit different and heterogeneous tissue properties depending on the tumor entity. Magnet resonance imaging (MRI) is one of the key elements in the diagnosis of primary brain tumors and in differentiating them from brain metastases or other intracranial lesions [1,2]. Discrimination between these different tumor entities is of high clinical importance as the tumors differ in prognosis, therapeutic approach and response due to diverging radio- and chemosensitivity [1]. As early treatment is associated with better clinical outcome, a fast, non-invasive differentiation of tumor entities would be desirable. So far, conventional MRI techniques, such as T2-weighted (w) imaging, fluid-attenuated inversion recovery (FLAIR) imaging, and gadolinium contrast-enhanced T1-weighted (w) imaging, do not allow for a distinct differentiation between tumor entities. Therefore, in general histopathological evaluation of the tumor via biopsy is necessary.

MRI is also used to evaluate the outcome of therapy for monitoring of treatment response. In general, the response is determined by changes in tumor volume according to the RANO (Response Assessment in Neuro-Oncology) criteria [3]. However, differentiation between tumor progression and pseudoprogression from radiochemotherapy remains challenging as sensitivity and specificity of current advanced imaging techniques such as diffusion-weighted imaging (DWI) or perfusion-weighted imaging (PWI) are limited [4].

Chemical exchange saturation transfer (CEST) is a novel non-invasive imaging technique based on magnetization transfer from protons in biomolecules to water [5,6,7]. This process can be induced in human tissue through frequency-selective saturation pulses and leads to a measurable reduction in the water signal. Consequently, the technique allows for non-invasive detection of designated low-concentrated proteins in tissue that cannot be visualized with conventional MRI [8,9,10,11]. Important target imaging molecules for CEST MRI contrasts are amide protons in mobile proteins and peptides. They exhibit a prominent signal at a frequency offset of approximately 3.5 ppm (with respect to the water signal) making them appropriate for detection in CEST imaging. The most widely used is amide proton transfer-weighted (APTw) imaging based on the asymmetry magnetization transfer ratio (MTRasym) analysis assuming that the MTRasym is that of magnetization transfer contrast effects, in contrast to CEST effects, being symmetrical about water [7]. Previous studies have shown that the concentration of amide protons is typically high in tumor lesions due to the high protein synthesis in fast proliferating tissues [8,12]. Notably, there is a fast-growing body of evidence in the literature for clinical applications of CEST imaging, especially in brain tumor imaging. APTw CEST imaging may aid in brain cancer diagnosis, such as the detection and grading of tumors [13,14,15], the assessment of therapy response and prognostication [16,17,18,19,20], and the identification of genetic biomarkers [21,22,23].

This prospective study aimed at applying APTw CEST imaging in a clinical setting in order to evaluate its diagnostic potential in differentiation between a brain tumor and normal-appearing white matter (NAWM), between a primary brain tumor and brain metastasis, as well as between patients with and without ongoing oncologic therapy. Results were compared to advanced imaging techniques, namely DWI with apparent diffusion coefficient (ADC) maps and PWI.

## 2. Materials and Methods

### 2.1. Patients

In this prospective study, 27 patients with intracranial tumors were included (six glioblastomas, four lymphomas, six meningiomas und 11 brain metastases). The primary tumors of the 11 patients with brain metastases were lung cancer, malignant melanoma, breast cancer and intestinal tumors; 17 patients received oncologic treatment at the time of image acquisition; 10 patients received no therapy defined as no oncological treatment at least 3 months prior to image acquisition. Inclusion criteria contained a histopathological confirmed intracranial tumor, visible contrast-enhancing tumor with a minimum diameter of 10 mm, age 18 years or older, and eligibility for clinical 3.0 T MRI. Demographics and further characteristics are summarized in Table 1. The study was approved by the local institutional ethics committee. All participants provided written informed consent.

### 2.2. Magnetic Resonance Imaging (MRI) Acquisition

Image acquisition was performed on a 3T clinical MRI scanner (MAGNETOM SKYRA 3T; Siemens Healthineers, Erlangen, Germany) including the following sequences: with and without gadolinium contrast-enhanced T1-w, T2-w imaging, FLAIR and DWI with generating apparent diffusion coefficient (ADC) maps. For five glioblastoma patients dynamic susceptibility contrast-enhanced MRI after gadolinium injection was acquired and relative cerebral blood volume (rCBV) maps were calculated.

For 3D CEST scan, interleaved saturation was realized by Gaussian-shaped radiofrequency pulses of a mean amplitude B1 = 1.2 μT, 99.8 ms length, 31 ms interpulse delay and with five pulses. Z-spectra were sampled at 16 regular frequency offsets (−300, ±4.97, ±4.26, ±3.55, ±2.84, ±2.13, ±1.42, ±0.71, 0). Images were acquired using a 3D centric reordered gradient-echo acquisition with 274 × 350 × 48 mm^3^ FOV, 128 × 128 × 16 matrix, 1.8 × 1.8 × 3.0 mm^3^ resolution, Grappa acceleration of 2, echo time (TE) = 2.16 ms, repetition time (TR) = 6.8 ms, 610 Hz/pixel acquisition bandwidth, 12° flip angle and a distance factor of 0.2.

### 2.3. Chemical Exchange Saturation Transfer (CEST) Data Post-Processing and Analysis

For post-processing, the APTw CEST was calculated using a MATLAB based software package kindly provided by the German Cancer Research Center (DKFZ), Heidelberg. Herein, the asymmetry analysis was performed at 3.5 ppm to eliminate overlapping effects of the direct water saturation and non-specific MRI effects. The MRI asymmetry signal at 3.5 ppm was calculated as the difference between the signal at −3.5 ppm (S-3.5 ppm) and +3.5 ppm (S3.5 ppm) divided by the unsaturated signal (S0). Z-spectra shifted according to the minimum in the Z-spectra after spline interpolation. All acquired image data were co-registered in each patient using the Medical Imaging Interaction Toolkit (MITK) created by the German Cancer Research Center [24].

The contrast-enhancing tumor area was manually segmented (by J.R., with 5 years of experience) under consideration of the hypointense areas on ADC map. Hemorrhagic tumor areas, tumor edema, necrosis or cystic tumor proportions were excluded. Additionally, in each patient a similar sized region of interest contralateral to the tumor site was selected in the NAWM. Mean MTRasym and 90th signal percentile MTRasym (90th pc MTRasym) were determined for the two regions of interest (tumor region and NAWM). For analysis of ADC maps and rCBV, the same voxels have been included.

### 2.4. Statistics

Statistical analysis was performed with SPSS Version 26 (IBM, Germany). To evaluate the diagnostic potential of APTw CEST for differentiation between tumor signal vs. NAWM signal, primary brain tumors vs. brain metastases and patients with therapy vs. without therapy, receiver operating characteristic (ROC) and area under the curve (AUC) analyses were performed. Results were compared with ADC maps and rCBV maps. *p* < 0.05 was considered to be statistically significant.

## 3. Results

### 3.1. Differentiation between Tumor Tissue versus Normal-Appearing White Matter (NAWM)

Mean MTR_asym_ and 90th pc MTR_asym_ revealed a statistically significant differentiation between tumor tissue of the whole tumor sample and contralateral NAWM yielding a sensitivity and specificity up to 89% and 78%, respectively (AUC = 0.82; *p* < 0.01) (Figure 1; Table 2). Of note, no statistically significant effect was found in the analysis of the ADC metrics.

Following subgroup analyses of each tumor entity vs. NAWM, statistically significant effects were found for the subgroups of glioblastomas (sensitivity/specificity: 100%/80%; AUC = 0.96; *p* < 0.01) (Figure 2) and meningiomas (100%/100%; AUC = 1.0; *p* < 0.01) but not for lymphomas (100%/50%; AUC = 0.75; *p* = 0.18) and metastases (73%/82%; AUC = 0.67; *p* = 0.2). In comparison, ADC metrics showed significant distinction in the subgroups of glioblastomas (100%/80%; AUC = 0.84; p = 0.02), metastases (82%/73%; AUC = 0.84; *p* = 0.04) and lymphomas (100%/100%; AUC = 1.0; *p* < 0.01). rCBV analysis showed a similar high performance to APTw CEST metrics in order to differentiate between glioblastomas and NAWM (100%/100%; AUC = 1.0; *p* < 0.01).

### 3.2. Differentiation between Primary Brain Tumor vs. Brain Metastases

Differentiation between primary brain tumors and metastases was possible with mean MTR_asym_ and 90th pc MTR_asym_ revealing up to 88% sensitivity and 63% specificity, respectively (AUC = 0.72; *p* = 0.04) (Figure 1; Table 2). Intriguingly, MTR_asym_ metrics for the NAWM showed no significant differences between primary brain tumors and brain metastases, respectively. No significant effects could be detected between the two groups using ADC metrics but ADC metrics tended to be more decreased in primary brain tumors than in brain metastases, possibly caused by the higher tumor cellularity of glioblastoma and lymphoma in the group of primary brain tumors.

### 3.3. Differentiation between Patients with vs. without Ongoing Oncological Therapy

Only the analysis of 90th pc MTR_asym_ demonstrated a significant difference between the group of patients with ongoing oncological therapy and the patients without therapy (Table 2). The sensitivity and specificity amounted to 80% and 53%, respectively (AUC = 0.71; *p* = 0.049). The mean and 10th pc ADC metrics were lower in the group under treatment compared to the group with no therapy but ADC finally did not allow for statistically significant differentiation between the two groups.

## 4. Discussion

By use of APTw CEST MRI at 3T in routine clinical practice, we assessed human intracranial brain tumors and found MTRasym metrics to differentiate statistically significant differences within the whole tumor sample between tumor tissue and contralateral NAWM (*p* < 0.01), and also in the subgroup comparisons of glioblastomas and meningiomas but not in lymphomas and metastases. Further analyses of the APTw CEST metrics demonstrated a significant difference between primary brain tumors and brain metastases as well as between the groups of patients with and without ongoing oncological therapy. In comparison, rCBV showed results comparable to MTRasym in glioblastomas, and ADC metrics revealed solely significant differences in the subgroups of glioblastomas, metastases and lymphoma.

### 4.1. Brain Tumor Tissue vs. NAWM

This study aimed to explore the application of APTw CEST imaging in a clinical setting in order to evaluate its diagnostic potential in differentiation between intracranial brain tumor and NAWM. Therefore, we investigated quantitatively MTR_asym_ metrics in human brain tumors compared to NAWM being a common approach in assessing CEST. Ma and colleagues [25] investigated the potential of CEST to distinguish pseudoprogression from true progression in malignant gliomas and reported CEST hyperintensity in true progression compared to NAWM. Moreover, Jiang et al. [19] found also high sensitivity and specificity using CEST to identify recurrent glioma taking the contralateral NAWM into account. In line with both previous studies and given that a high APTw CEST signal is associated with increased amide proton concentrations, the findings of our study supported the excellent ability to differentiate between glioblastomas and NAWM by CEST.

Through methodological stringency, the further subgroup analyses in this study revealed significant differences between tumor tissue and NAWM in meningiomas but not in lymphomas and metastases. For meningioma, Joo et al. [15] and Yu et al. [14] found significantly decreased CEST signal in meningioma WHO Grade I compared to meningioma WHO Grad II. Due to the small sample size in our cohort of six meningiomas a further subdivision was not feasible. However, the current study let us assume that CEST differentiates between tissue of meningioma and NAWM. For lymphomas and metastases in contrast, Jiang et al. [13] found reduced CEST metrics of lymphomas compared to high-grade gliomas, and Kimberly et al. [26] reported that CEST metrics predicted response to stereotactic radiosurgery in human brain metastasis. In particular, the latter study observed merely a uniform therapy concept whereas in the current study most patients received a comprehensive combination therapy including surgery and/or radiochemotherapy, possibly therefore resulting in the lack of significance. In addition to the potential effect of treatments and according to a previous report of Mehrabian and colleagues [27] that CEST in contralateral NAWM of glioblastoma was different compared to healthy controls, it should also be considered that in the present study CEST metrics of contralateral NAWM in lymphomas and metastases were higher than in meningiomas. Even though highly speculative, it might be considered whether the lack of significance in the subgroup analysis of lymphomas and metastases with NAWM was possibly caused by modified brain metabolism due to tumor cell infiltration.

In comparison with APTw CEST, rCBV metrics showed similar high results in distinguishing glioblastoma from NAWM, in line with previous findings [28] and therefore underpinning the capacity of our findings. However, the absence of a significant ADC decrease in lymphomas is uncommon [29] and might be explained by treatment effects [30]. This aspect must carefully be considered when evaluating our CEST metrics, even though ADC metrics tended to be decreased in lymphomas compared to NAWM. As it is known that ADC metrics can be decreased in glioblastomas as well as brain metastases, it might not be surprising that statistical analyses showed significant results there.

### 4.2. Primary Brain Tumors vs. Brain Metastases

Performing ROC analyses, MTR_asym_ metrics showed good potential in distinguishing between primary brain tumors and metastases yielding sensitivity of 88% and specificity of 63% at a statistically significant level (*p* < 0.04). In a cohort of 31 patients with glioblastoma und 17 patients with brain metastases Kamimura et al. [31] derived from a histogram analysis significantly higher CEST metrics in glioblastomas compared to metastases. Although in the present study group analysis was performed between metastases and the whole primary brain tumor sample, there might be an obvious comparability to our study as glioblastomas presented the group with the highest CEST metrics of the whole tumor sample and thus glioblastomas possibly contributed most to the observed effects. Therefore, in line with the literature, the present study supported again CEST to be helpful in distinguishing between primary brain tumors and metastases. Its importance becomes obvious considering that treatment strategy can substantially differ between both entities not only in adults but also in children [32]. As expected, ADC maps showed no significant difference between primary brain tumors and metastases [33,34].

### 4.3. Patients with vs. without Ongoing Oncological Therapy

Analysing APTw CEST metrics between patients with vs. without ongoing oncological treatment revealed significant differences for 90th pc MTR_asym_ but not for mean MTR_asym_. As already discussed above, previous studies in glioblastomas also reported that CEST might be capable to portray treatment effects [2,17,25,26,35,36,37]. The underlying mechanism behind CEST signals in brain tumors remains an interesting research topic. It is assumed that CEST allows an indirect quantification of cellular proliferation and the identification of active tumor parts [38]. The findings of our study confirmed the potential of CEST to observe treatment effects in a clinical setting even on a probable subtle level considering that even the group without ongoing treatment might have received therapy more than 3 months before MRI acquisition.

### 4.4. Limitations

There are some limitations that need to be acknowledged. Firstly, we restricted the outlined region of interest to the solid contrast-enhancing tumor area being a comprehensible imaging feature unifying the highly varied investigated tumor sample. Future studies with a more homogenous study sample should also address non contrast-enhancing tumor parts since effects have also been reported there [35]. Secondly, in several cases wasn’t tumor status histopathologically confirmed by biopsy and, thus, some false positive subjects might have been included. On the one hand in our clinic evaluation of tumor status is assessed by a highly professional multidisciplinary team reducing the rate of false positive assignment to its lowest and on the other hand a high false positive rate would have reduced the levels of significance which, however, we did not observe. Moreover, an explorative analysis with exclusion of these cases did not significantly influence the findings of the present study. Thirdly, given the comparison of primary brain tumors vs. brain metastases revealed significant differences, in this study subgroup comparisons at the level of each tumor entity for therapy status was not feasible due to the small sample size. The findings of our explorative analysis within the whole tumor sample, however, might nevertheless provide clues in this field for future studies with a larger patient cohort. Finally, there is no standard CEST sequence making it in general complicated comparing CEST metrics between different studies. In the present study, the sequence parameters were chosen for asymmetry analysis to quantify selective amide signal which up to now has been one of the most common approaches in CEST imaging. However, it is known that despite the use of asymmetry analysis, certain overlays over the absolute amide-signal remain, caused by the relayed nuclear Overhauser effect and the semisolid magnetization transfer [36]. Therefore, newer imaging approaches have been proposed such as the AREX (apparent exchange-dependent relaxation) method [39,40]. Recently, the so called ‘Pulseq-CEST project’ has been launched for sharing current sequence parameters and for developing an open-source sequence standard [41].

## 5. Conclusions

This study confirmed the diagnostic potential of APTw CEST imaging in a clinical setting in order to differentiate in vivo between brain tumor tissue and NAWM, between primary brain tumors and brain metastases, as well as between tumor tissue with and without ongoing oncologic therapy. Therefore, the results of our study may support the use of CEST as a non-invasive in vivo imaging biomarker for categorization between tumor entities and monitoring treatment response in routine clinical practice.

## Figures and Tables

**Figure 1 diagnostics-12-00490-f001:**
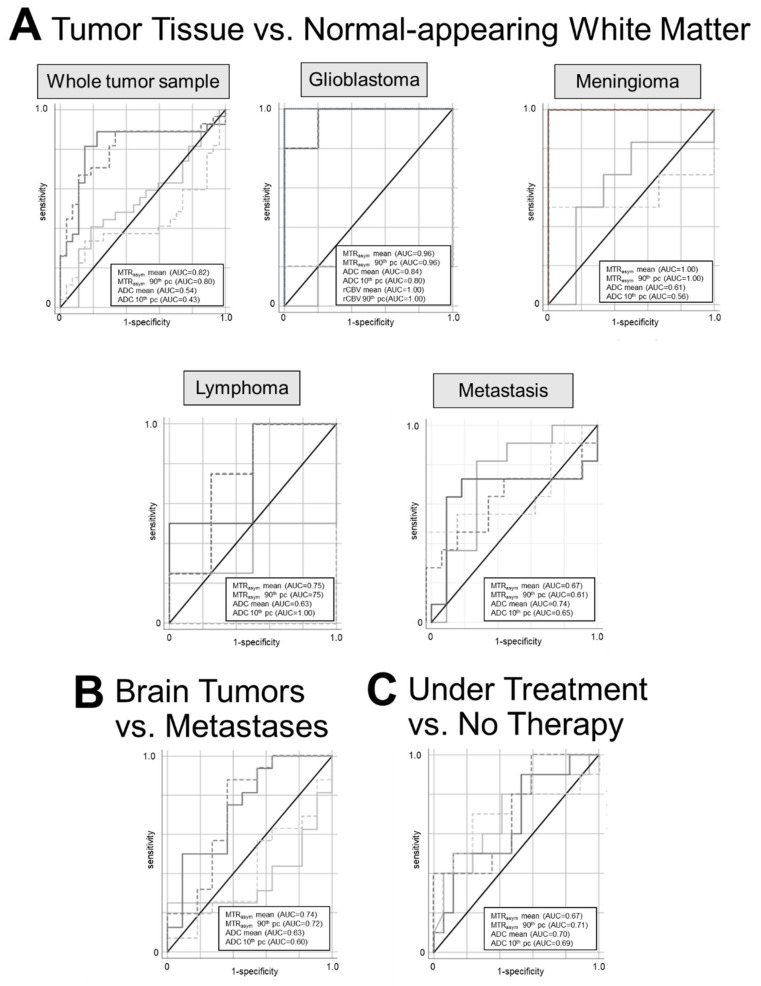
Receiver operating curve characteristics (ROC) curves and area under the curve (AUC) analyses for mean asymmetry magnetization transfer ratio (MTRasym mean: solid dark grey line), 90th percentile MTRasym (MTRasym 90th pc; dashed dark grey line), apparent diffusion coefficient mean (ADC mean; solid light grey line), 10th percentile ADC (ADC 10th pc; dashed light grey line), relative cerebral blood volume mean (rCBV mean; dotted black line), 90th pc rCBV (rCBV 90th pc; dotted grey line). AUC values are displayed in brackets. (**A**) Predictability of tumor tissue vs. normal-appearing white matter in the whole tumor sample and subgroups (glioblastoma multiforme, meningioma, lymphoma, intracerebral metastasis). (**B**) Differentiation between primary brain tumors and metastases. (**C**) Differentiation between patients with and without ongoing oncologic treatment.

**Figure 2 diagnostics-12-00490-f002:**
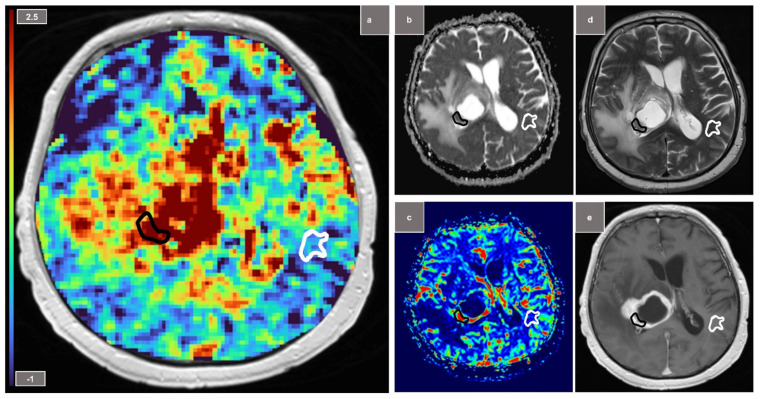
Glioblastoma with tumor ROI (region of interest) marked in black and NAWM (normal-appearing white matter) ROI outlined in white. (**a**) Amide proton transfer-weighted chemical exchange saturation transfer (APTw CEST) map (**b**) ADC (apparent diffusion coefficient) map (**c**) relative cerebral blood volume (rCBV) map (**d**) T2-weighted TSE (**e**) gadolinium contrast-enhanced T1-weighted image.

**Table 1 diagnostics-12-00490-t001:** Patients’ characteristics.

	Number	Age (Mean + SD)	Sex (m/f)	Under Treatment vs. No Therapy
Glioblastoma	6	60.50 ± 4.23	5/1	4 vs. 2
Lymphoma	4	74.50 ± 5.69	1/3	3 vs. 1
Meningioma	6	59.00 ± 18.47	2/4	2 vs. 4
Metastases	11	61.82 ± 6.69	3/8	8 vs. 3
All	27	62.48 ± 10.77	11/16	17 vs. 10

Tumor entity, age at the time of image acquisition, sex, therapy status. SD = standard deviation; m = male; f = female.

**Table 2 diagnostics-12-00490-t002:** Results of receiver operating characteristic analyses of MTRasym, ADC and rCBV at 3T CEST magnetic resonance imaging (MRI) in routine clinical practice.

	Comparison of Means	Sensitivity/Specificity	AUC/Cut-Off	*p*-Value
**A Tumor Tissue vs. Normal-Appearing White Matter (NAWM)**				
**Whole tumor sample vs. NAWM**				
**MTR_asym_ mean**	0.32 vs. −0.64	0.89/0.78	0.82/−0.22	**<0.01**
**MTR_asym_ 90th pc**	2.27 vs. 0.77	0.89/0.67	0.80/0.96	**<0.01**
**ADC mean**	932.71 vs. 896.71	0.56/0.52	0.54/860.18	0.60
**ADC 10th pc**	709.21 vs. 729.55	0.59/0.27	0.43/678.50	0.36
**Glioblastoma vs. NAWM**				
**MTR_asym_ mean**	0.79 vs. −1.09	1.00/0.80	0.96/−0.52	**<0.01**
**MTR_asym_ 90th pc**	2.22 vs. 0.1	1.00/0.80	0.96/0.56	**<0.01**
**ADC mean**	846.59 vs. 974.92	1.00/0.80	0.84/828.46	**0.02**
**ADC 10th pc**	777.12 vs. 869.75	1.00/0.80	0.80/726.35	0.09
**rCBV_mean_**	771.77 vs. 294.86	1.00/1.00	1.00/465.24	**<0.01**
**rCBV_90_**	1174.12 vs. 476.88	1.00/1.00	1.00/897.75	**<0.01**
**Lymphoma vs. NAWM**				
**MTR_asym_ mean**	0.70 vs. −0.22	1.00/0.50	0.75/−0.41	0.18
**MTR_asym_ 90th pc**	3.07 vs. 1.52	1.00/0.50	0.75/0.96	0.18
**ADC mean**	896.57 vs. 884.56	1.00/0.50	0.63/798.87	0.58
**ADC 10th pc**	540.25 vs. 710.63	1.00/1.00	1.00/640.75	**<0.01**
**Meningioma vs. NAWM**				
**MTR_asym_ mean**	0.54 vs. −0.66	1.00/1.00	1.00/−0.01	**<0.01**
**MTR_asym_ 90th pc**	3.38 vs. 0.68	1.00/1.00	1.00/1.63	**<0.01**
**ADC mean**	963.41 vs. 902.76	0.67/0.66	0.61/864.41	0.53
**ADC 10th pc**	685.7 vs. 697.67	0.50/0.50	0.56/713.00	0.77
**Metastasis vs. NAWM**				
**MTR_asym_ mean**	−0.2 vs. −0.53	0.73/0.82	0.67/−0.22	0.2
**MTR_asym_ 90th pc**	1.4 vs. 0.92	0.73/0.55	0.61/0.95	0.38
**ADC mean**	976.07 vs. 855.16	0.82/0.73	0.74/847.31	**0.04**
**ADC 10th pc**	746.43 vs. 677.35	0.55/0.81	0.65/745.60	0.25
**B Primary Brain Tumors vs. Metastases**				
**MTR_asym_ mean**	0.68 vs. −0.2	0.81/0.55	0.74/0.1	**0.02**
**MTR_asym_ 90th pc**	2.86 vs. 1.4	0.88/0.63	0.72/1.54	**0.04**
**ADC mean**	902.19 vs. 976.07	0.64/0.69	0.63/923.33	0.26
**ADC 10th pc**	683.62 vs. 746.43	0.55/0.75	0.60/734.40	0.40
**C Under Treatment vs. No Therapy**				
**MTR_asym_ mean**	0.07 vs. 0.74	0.60/0.53	0.67/0.26	0.12
**MTR_asym_ 90th pc**	1.73 vs. 3.17	0.80/0.53	0.71/1.66	**0.049**
**ADC mean**	887.27 vs. 1009.95	0.70/0.59	0.70/864.98	0.11
**ADC 10th pc**	671.24 vs. 773.75	0.60/0.77	0.69/734.40	0.12

asymmetry magnetization transfer ratio (MTRasym), apparent diffusion coefficient (ADC), relative cerebral blood volume mean (rCBV), chemical exchange saturation transfer (CEST), area under curve (AUC), normal-appearing white matter (NAWM); percentile (pc).

## Data Availability

The data presented in this study are available upon request from the corresponding author.

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
