# Peer review of "The Value of APTw CEST MRI in Routine Clinical Assessment of Human Brain Tumor Patients at 3T"

_diagnostics, 2022, doi:10.3390/diagnostics12020490_

Round 1
Reviewer 1 Report
Manuscript ID: Diagnostics-1592192
Type of manuscript: Article
Title: The value of APTw CEST MRI in routine clinical assessment of human brain tumor patients at 3T.
The authors conducted a prospective study to evaluate the diagnostic potential in the differentiation of intracranial tumors applying Amide Proton Transfer weighted (APTw) CEST imaging in a clinical setting. They employed chemical exchange saturation transfer (CEST) magnetic resonance imaging (MRI) for comparing tumor areas and similar-sized normal regions on the contralateral hemisphere in 27 patients with intracranial tumors. The CEST signals were analyzed in the tumor areas and contralateral normal-appearing white matter (NAWM) using the asymmetry magnetization transfer ratio (MTRasym).
The results indicated significantly higher contrast-enhancing tumor areas by APTw CEST metrics than contralateral NAWM. Interestingly, subgroup analyses of each tumor entity versus NAWM showed statistically significant results for glioblastomas and meningiomas but not for lymphomas or metastases. Besides, perfusion-weighted imaging (PWI) showed results comparable to APTw CEST in glioblastoma, supporting the diagnostic usefulness of APTw CEST imaging in a routine clinical setting to differentiate brain tumors.
The investigation's objectives and results are accurately presented and discussed, making the manuscript suitable for publication as a proof of concept. However, there are a few clarifications the authors have to address.
- Interestingly, MTRasym metrics for the NAWM showed no significant differences between primary brain tumors and brain metastases, respectively. However, ADC metrics tended to be more decreased in primary brain tumors than in brain metastases. Though no significant effects were detected between the two groups using ADC metrics, ADC metrics tended to be more decreased in primary brain tumors than in brain metastases. Do the authors have any explanation for these results?
- Since a high APTw CEST signal is associated with increased amide proton concentrations, which correlates with high mobile proteins and peptides such as protein synthesis in fast proliferating tissues, it is surprising that the analysis of the subgroup tumors reveals significant differences between tumor tissue and NAWM in meningiomas but not in lymphomas and metastases. Usually, lymphomas and metastatic carcinomas have higher metabolic rates than meningiomas. Do the authors have any pathophysiological explanation in addition to the potential effect of treatments? The mechanisms underlying CEST signals in brain tumors remain an exciting research issue. However, it is thought that CEST permits an indirect quantification of cellular proliferation labeling functional tumor regions. Are APTw CEST signals associated with NAWM in lymphoma and metastatic carcinomas patients higher than NAWM in meningioma patients?
- Regarding the study's limitations, the authors indicated that not in all cases the tumor status was histopathologically confirmed by biopsy and, therefore, some false positive cases might have been included. Are the results significantly different if those few cases were excluded?
Author Response
Point 1: Interestingly, MTRasym metrics for the NAWM showed no significant differences between primary brain tumors and brain metastases, respectively. However, ADC metrics tended to be more decreased in primary brain tumors than in brain metastases. Though no significant effects were detected between the two groups using ADC metrics, ADC metrics tended to be more decreased in primary brain tumors than in brain metastases. Do the authors have any explanation for these results?
Response 1: We thank the reviewer for addressing this interesting aspect. The group of primary brain tumors comprises of GBM and Lymphoma which are associated with high cellularity and therefore maybe yielding more decreased ADC metrics than brain metastes. As there is some overlap between ADC metrics in tumors of the brain results are not statistically significant.
This aspect was added to the results section: “(…) ADC metrics tended to be more decreased in primary brain tumors than in brain metastases, maybe caused by the higher tumor cellularity of glioblastoma and lymphoma in the group of primary brain tumors.”
Point 2: Since a high APTw CEST signal is associated with increased amide proton concentrations, which correlates with high mobile proteins and peptides such as protein synthesis in fast proliferating tissues, it is surprising that the analysis of the subgroup tumors reveals significant differences between tumor tissue and NAWM in meningiomas but not in lymphomas and metastases. Usually, lymphomas and metastatic carcinomas have higher metabolic rates than meningiomas. Do the authors have any pathophysiological explanation in addition to the potential effect of treatments? The mechanisms underlying CEST signals in brain tumors remain an exciting research issue. However, it is thought that CEST permits an indirect quantification of cellular proliferation labeling functional tumor regions. Are APTw CEST signals associated with NAWM in lymphoma and metastatic carcinomas patients higher than NAWM in meningioma patients.
Response 2: We thank the reviewer for his/her valuable remark. We fully agree that CEST is thought to permit an indirect quantification of cellular proliferation labeling functional tumor regions. Our results indicated that CEST signals of NAWM in lymphomas and metastases were higher than in meningiomas (0.22 and -0.53, respectively > -0.66). Although it remains highly speculative it might be discussed that a modified brain metabolism due to tumor cell infiltration is also present in contralateral NAWM, in line with a previously report of Mehrabian and colleagues (Mehrabian al 2018) that CEST in contralateral NAWM of glioblastoma was different compared to healthy controls.
The following paragraph was added to the discussion section: “In addition to the potential effect of treatments and according to a previous report of Mehrabian and colleagues [27] that CEST in contralateral NAWM of glioblastoma was different compared to healthy controls, it should also be considered that in the present study CEST metrics of contralateral NAWM in lymphomas and metastases were higher than in meningiomas. Even though highly speculative it might be discussed that the lack of significance in the subgroup analysis of lymphomas and metastases with NAWM was maybe caused by modified brain metabolism due to tumor cell infiltration.”
Point 3: Regarding the study's limitations, the authors indicated that not in all cases the tumor status was histopathologically confirmed by biopsy and, therefore, some false positive cases might have been included. Are the results significantly different if those few cases were excluded?
Response 3: We thank the reviewer for the scientific advice. For an explorative analysis we excluded the three cases in the group of metastases and the results remained stable and did not differ significantly, e.g. whole tumor sample vs NAWM, p<0.01.
This information was added to the limitation paragraph and reads as follows: “Secondly, not in any case tumor status was histopathological confirmed by biopsy and, thus, some false positive subjects might have been included. On the one hand in our clinic evaluation of tumor status is assessed by a highly professional multidisciplinary team reducing the rate of false positive assignment to its lowest and on the other hand a high false positive rate would have reduced the levels of significance which, however, we did not observe. Moreover, an explorative analysis with exclusion of these cases did not significantly influence the findings of the present study."
Reviewer 2 Report
The article is devoted to the current topic - the study of the possibilities of instrumental research methods in diagnosing brain tumors. The promising opportunities and advantages of the ways in a comparative aspect are revealed.
Author Response
The article is devoted to the current topic - the study of the possibilities of instrumental research methods in diagnosing brain tumors. The promising opportunities and advantages of the ways in a comparative aspect are revealed.
Response: We would like to thank the reviewer for his/her friendly comments.
Reviewer 3 Report
Innovative approach to studying brain tumors.
Exhaustive introduction where the context from which the study starts and the aims of scientific research are clearly described.
Appropriate methods.
Rigorous statistical analysis.
Clear and simple description of the results despite the fact that the diagnostic technique used is also quite complex.
The conclusions of the study are very interesting especially for the implication in the clinical care activity.
Author Response
Innovative approach to studying brain tumors. Exhaustive introduction where the context from which the study starts and the aims of scientific research are clearly described. Appropriate methods. Rigorous statistical analysis. Clear and simple description of the results despite the fact that the diagnostic technique used is also quite complex. The conclusions of the study are very interesting especially for the implication in the clinical care activity.
Response: We would like to thank the reviewer for his/her friendly comments.